# The Role of Mesenchymal Reprogramming in Malignant Clonal Evolution and Intra-Tumoral Heterogeneity in Glioblastoma

**DOI:** 10.3390/cells13110942

**Published:** 2024-05-30

**Authors:** Qiong Wu, Anders E. Berglund, Robert J. Macaulay, Arnold B. Etame

**Affiliations:** 1Department of Neuro-Oncology, H. Lee Moffitt Cancer Center and Research Institute, 12902 Magnolia Drive, Tampa, FL 33612, USA; 2Department of Biostatistics and Bioinformatics, H. Lee Moffitt Cancer Center and Research Institute, 12902 Magnolia Drive, Tampa, FL 33612, USA; 3Departments of Anatomic Pathology, H. Lee Moffitt Cancer Center and Research Institute, 12902 Magnolia Drive, Tampa, FL 33612, USA

**Keywords:** glioblastoma, intra-tumoral heterogeneity, mesenchymal reprogramming, clonal evolution, anti-GBM therapy, tumor microenvironment, therapeutic resistance

## Abstract

Glioblastoma (GBM) is the most common yet uniformly fatal adult brain cancer. Intra-tumoral molecular and cellular heterogeneities are major contributory factors to therapeutic refractoriness and futility in GBM. Molecular heterogeneity is represented through molecular subtype clusters whereby the proneural (PN) subtype is associated with significantly increased long-term survival compared to the highly resistant mesenchymal (MES) subtype. Furthermore, it is universally recognized that a small subset of GBM cells known as GBM stem cells (GSCs) serve as reservoirs for tumor recurrence and progression. The clonal evolution of GSC molecular subtypes in response to therapy drives intra-tumoral heterogeneity and remains a critical determinant of GBM outcomes. In particular, the intra-tumoral MES reprogramming of GSCs using current GBM therapies has emerged as a leading hypothesis for therapeutic refractoriness. Preventing the intra-tumoral divergent evolution of GBM toward the MES subtype via new treatments would dramatically improve long-term survival for GBM patients and have a significant impact on GBM outcomes. In this review, we examine the challenges of the role of MES reprogramming in the malignant clonal evolution of glioblastoma and provide future perspectives for addressing the unmet therapeutic need to overcome resistance in GBM.

## 1. Introduction

Glioblastoma (GBM) is the most common infiltrative primary central nervous system (CNS) malignancy with a median survival of 15 months [1,2,3]. While genetic and environmental factors have been postulated as contributory factors, the overwhelming majority of GBM cases are sporadic. Advances in GBM epidemiology have resulted in the appreciation of biological subtypes and also the relevance of these subtypes to GBM outcomes [4]. Nevertheless, there is a substantial unmet need for impactful therapeutic strategies that can significantly improve GBM outcomes beyond the standard of care. 

GBM standard therapy entails multimodal strategies of maximum surgical resection, temozolomide (TMZ) chemotherapy, and radiotherapy. However, tumor recurrence is universal and frequent. Tumor heterogeneity, the tumor infiltrative growth pattern, and the central nervous system location present significant challenges to current therapeutic approaches, leading to disease recurrence. As a consequence, therapeutic resistance remains a largely unaddressed phenomenon in GBM. Molecular surrogates of favorable GBM biology and therapeutic response have been proposed [5,6,7,8,9,10]. One of the most impactful genetic alterations that govern glioma tumor biology and permit clinically relevant classification is *IDH* genomic status [5,10]. The importance of the NADP(+)-dependent isocitrate dehydrogenases protein encoded by *IDH1* and *IDH2* genes has been known for over a decade, whereby *IDH1* mutations are present in high-grade gliomas that develop from low-grade gliomas, whereas *IDH*-wildtype GBMs arise de novo and usually have a poorer prognosis [5,10]. Another molecular prognosticating marker of chemotherapeutic response is the promoter methylation status of the DNA repair enzyme O6-methylguanine-DNA methyltransferase (MGMT) [7,8,9]. *MGMT* is epigenetically regulated in high-grade gliomas [8]. Epigenetic silencing of *MGMT* sensitizes GBMs to TMZ and improves survival, thereby rendering *MGMT* methylation status as one of the most important biomarkers to predict TMZ response [7]. Thus far, *MGMT* methylation status and mutation in *IDH1* are the most impactful independent prognosticating factors in the clinical management of GBM [6]. 

Beyond the standard of care treatment, targeted inhibition of crucial growth factor pathways [11], immune-checkpoint inhibitors, and tumor vaccine strategies [12,13,14,15,16] have been extensively explored in GBM for therapeutic efficacy. To date, none of the above strategies have been effective [13,17,18,19,20,21]. A combination of immunosuppressive microenvironment factors, as well as tumor molecular and cellular heterogeneity factors, have been postulated as contributors to therapeutic futility in GBM [22,23,24,25,26,27]. Furthermore, treatment-induced mesenchymal (MES) reprogramming has emerged as the leading cause of intra-tumoral heterogeneity, malignant clonal evolution, and subsequent lethality in GBM [28,29,30]. Hence, systematic characterization of malignant reprogramming mechanisms can provide valuable insights into novel therapeutic interventions in GBM. This review will highlight the role of MES reprogramming in therapeutic futility in glioblastoma and provide future perspectives for addressing this formidable challenge in GBM.

## 2. GBM Stem Cells in the GBM Microenvironment

Regional heterogeneity is both a histopathological and a radiographic hallmark of GBM, whereby there are regions of central hypoxia and necrosis surrounded by a pseudo-palisading, a proliferative angiogenic zone that is enhanced in contrast magnetic resonance imaging (MRI) [31]. Through bulk tumor analysis, it has been demonstrated that the GBM heterogeneous subclones evolve from a subset of stem-like cells known as the GBM stem cells (GSCs), which harbor distinct genetic alterations [32] and originate from neural stem cells of the subventricular neurogenesis zone [33,34]. GSCs have self-renewal capabilities and are characterized by evaluating the expression of specific gene markers reflective of stemness including CD133, Sox2, and Nestin [35,36] (Figure 1). In light of their high proliferation rate and molecular heterogeneity, GSCs are highly resistant to GBM therapy and serve as a critical nidus for disease recurrence [37,38]. Interestingly, the GSCs within the perinecrotic hypoxic niche and angiogenic niche are highly proliferative, relative to GSCs of the brain-invasive front [39]. Post-treatment recurrence is believed to be secondary to the repopulation of new tumors by GSCs that persist despite treatment [40]. Therefore, there is overwhelming evidence that GSCs are the primary contributor to tumorigenicity, treatment-induced resistance, and recurrence. Furthermore, there is evidence that GSCs mediate therapeutic resistance through multiple mechanisms, impacting DNA repair and drug efflux systems [41,42] (Figure 1).

Given the apical location of GSCs within the GBM cellular hierarchy, GCSs play a critical role in GBM tumorigenicity and cell fate determination [39,43,44]. GSCs co-exist with differentiated tumor cells, astrocytes, and immune cells within the perivascular niche of GBM [39]. Complex interactions within this perivascular microenvironment sustain GSC survival and proliferation. Immune cells within the tumor microenvironment include tumor-associated macrophages (TAMs), microglia, myeloid-derived suppressor cells (MDSCs), neutrophils, and monocytes [39,45]. TAMs and microglia play vital roles in GSC tumorigenesis through the upregulation of matrix metalloproteinase 9 (MM9) expression via transforming growth factor-β (TGF-β) signaling [46,47]. Furthermore, the maintenance of GSC self-renewal is sustained through tumor cell-induced paracrine proliferation and migration of astrocytes [48,49]. 

## 3. Heterogeneity in GBM

Intra-tumoral heterogeneity, both at the cellular and molecular genetics level (Table 1), is a pathognomonic hallmark of GBM that is responsible for therapeutic resistance and poor outcomes in GBM [27]. Hence, a deeper understanding of the nature of cellular and molecular heterogeneity in GBM is essential to developing therapies that are impactful in GBM. Over the last two decades, there have been significant advances in deciphering critical genetic alterations in GBM, which have facilitated both tumor characterization and an enhanced appreciation of the GBM landscape. The Cancer Genome Atlas (TCGA) was the seminal study that characterized critical molecular pathway aberrations that were highly featured in GBM through the comprehensive characterization of over 600 GBM patient tissues [50]. In particular, mutations in *TP53*, receptor tyrosine kinase genes (RTKs), and *RB* were identified as the most common critical genetic alterations in GBM [51]. Loss of function of the tumor suppressor gene *TP53* through mutation or alterations of other p53 signaling components such as MDM2 promotes the malignant reprogramming of tumor cells [52,53,54]. RB signaling, which is tumor suppressive, is highly dysregulated in GBM through aberrations of crucial activators of p53 such as CDK4 amplification and CDKN2A deletion in GBM [55]. In addition, pervasive alterations to RTK signaling pathways, including EGFR, PDGF, and TGF-β, facilitate GBM oncogenesis through the downstream activation of oncogenic pathways [39]. Signal transduction through RTKs facilitates important oncogenic processes at the cellular level including proliferation, apoptosis resistance, and invasion [56,57]. Amplification of EGFR as well as constitutive-active mutants (EGFRvIII) represents the most common alteration in GBM [58,59]. Similarly, PDGFRA amplification [60] and deletional mutations [61] are commonly encountered in GBM. Downstream activations of RAS/MAPK and PI3K/AKT/mTOR signaling pathways through mutations and deletions of pathway components appear to be common oncogenic and malignant propagating events in GBM [60]. The diverse activations and complex interactions of multiple oncogenic signaling pathways in GBM are key aspects of GBM heterogeneity that present unique therapeutic challenges.

Molecular interactions between tumor cells and non-tumor cells within the GBM tumor microenvironment (TME) add further complexity to intra-tumoral heterogeneity. Fortunately, the Ivy Glioblastoma Atlas Project (https://glioblastoma.alleninstitute.org/, accessed on 22 May 2024) made significant contributions toward understanding the genetic landscape in GBM from a regional perspective (Table 1). Biopsy samples were obtained using image-guided investigation of MRI-distinct regions in GBM, and the tissue was subjected to bulk RNA sequencing (RNA-seq). There were marked variations and significant regional heterogeneity based on the top 10 gene expression profiles (Table 1). Although the bulk tumor data were quite valuable, molecular details at the cellular level are more informative of intra-tumoral heterogeneity. Recent advances in single-cell RNA sequencing (scRNA-seq) have bridged the gap between the molecular profiling features of a bulk tumor and individual tumor cells. For instance, within the bulk tumor, there are variations in individual tumor cell gene expression and clustering, leading to heterogeneity in the GBM molecular subtype profiles of individual tumor cells with implications for therapeutic resistance [27]. Furthermore, scRNA-seq investigations have implicated the TME as a malignant facilitator of GBM through reprogramming mechanisms involving hypoxia [62], immunosuppression [63,64], MES reprogramming [65,66], and cellular metabolism [67,68]. Most recently, the emergence of spatial transcriptomics has permitted a deeper investigation of cellular interactions within the TME and optimal delineation of cellular niches within the tumor [69,70,71]. In a very recent study, Greenwald et al. defined GBM cellular states and uncovered their organization through approaches combining spatial transcriptomics, spatial proteomics, and computational analysis [72]. Their findings indicated that GBM tumors contain both disorganized and structured regions, whereby the organized regions were associated with an abundance of MES-hypoxic cancer cells that extended beyond what could be observed in histopathology. 

The clonal evolution of GSCs and non-GSC populations and subsequent interactions with the tumor microenvironment (TME) contribute to heterogeneity. GSCs are constantly in a state of equilibrium toward differentiation into non-GSC populations versus the maintenance of stemness. Stemness hierarchical plasticity is the basis for initiating the recurrent tumor after cytotoxic therapy. In terms of cellular architecture heterogeneity, GSC subpopulations can be classified as oligodendrocyte progenitor cells, neural progenitor cells, astrocyte-like cells, or mesenchymal-like cells [73,74]. It is now apparent that *IDH* mutation status influences both the GSC proliferation state and cellular architecture [75]. For instance, the GSCs in *IDH*-mutant tumors are in a non-proliferative state compared to GSCs in *IDH*-wild-type tumors where GSCs are highly proliferative. Furthermore, while *IDH*-mutant and *IDH*-wild-type tumors consist of mixed GSC subpopulations, the proliferating mesenchymal-like cells are most commonly associated with *IDH*-wild-type GBM [75]. 

Further insights into the molecular inter-tumoral and intra-tumoral heterogeneities of GSCs have emerged secondary to large-scale genomic and RNA sequencing investigations, that reveal GBM segregation into distinct survival prognostic molecular subtypes [76,77]. Using a combination of gene expression, mutational, and copy number analysis, Verhaak et al. subsequently stratified GBM into the following four distinct molecular subtypes reflecting inter-tumoral heterogeneity: proneural (PN), mesenchymal (MES), neural (NL), and classical (CL) [51]. PN tumors are often enriched in oligodendrocytic signature, have the best prognosis, and are characterized by mutations in *PDGFRA* and *IDH1/2* [51]. MES subtype tumors have a strong astrocytic signature, have the worst prognosis, and are genetically characterized by *NF1* mutations [51]. The classical subtype has an astrocytic signature as well but is characterized by *EGFR* aberrations [51]. The neural subtype has both astrocytic and oligodendrocytic signatures and is characterized by neuronal gene expressions [51]. Although the molecular subtypes in GBM have been identified based on individual tumor analysis, it is now evident from the stereotactic surgery investigation of different GBM regions in a single patient that more than one subtype can exist within the same tumor [78]. Intra-tumoral heterogeneity in GBM molecular subtypes has significant clinical implications with respect to therapeutic response and prognosis. 

A unique advantage of the classification of GBM into distinct molecular subtypes is the association of genetic heterogeneity with therapeutic resistance and tumor recurrence. For instance, the single-cell RNA sequencing (scRNA-Seq) of GBM further reveals the impact of intra-tumoral heterogeneity in molecular subtypes on GBM survival [27]. It became evident that all GBM tumors have PN subpopulations and that it was the variance of PN subpopulations relative to the other molecular subtypes that impacts survival [27]. MES subpopulations are highly resistant to therapy and confer dismal survival compared to other subtypes. TCGA dataset analysis revealed *NF1* mutations and NF-κB signaling aberrations as facilitators of the MES subtype [79]. Furthermore, TNF-α/NF-κB signaling drives radiation resistance in GBM through the PN to MES transition of GBM stem cells [80]. Similarly, the induction of TGF-β2 through the dephosphorylation of OLIG2 facilitates MES transition [81]. Hence, within the context of molecular intra-tumoral heterogeneity, MES proclivity significantly impacts survival and contributes to variations in clinical outcomes. 

## 4. MES Reprogramming 

MES reprogramming is a cellular process during which cancer cells acquire enhanced migratory and invasive characteristics contributing to malignant transformation and propagation [82]. MES reprogramming is driven by signaling networks involving transcriptional factors and downstream effectors, and reprogramming is often the aftermath of interactions between cancer cells and the TME or therapeutic exposure [82]. Although MES reprogramming was traditionally considered as a phenomenon mainly unique to epithelial cancers, the MES state of GBM has been identified through molecular clustering whereby *NF1* loss appears to be a consistent genetic lesion [51]. There is mounting evidence that therapeutic resistance and recurrence in GBM are associated with enhanced MES phenotype reprogramming [83,84,85]. For instance, detailed analyses of recurrent GBMs have uncovered evidence of molecular subtype transitions as the basis for chemotherapy and radiotherapy resistance [77,85,86,87,88]. PN towards MES reprogramming represents the most common molecular subtype transition whereby PN genes are down-regulated and MES genes are upregulated [80,89,90,91]. MES and PN preclinical genetic models of GBM driven by *NF1* loss and *PDGFB* overexpression, respectively, demonstrate differential responses to radiation (RT) and TMZ whereby the *PDGFB* overexpression phenotype is more sensitive compared to the *NF1* loss phenotype [92]. Interestingly, therapy-resistant GBMs have an MES-like phenotype, while therapy-sensitive GBMs have a PN-like phenotype [85]. Given the profound negative impact of MES reprogramming on GBM outcomes, there is urgency and renewed emphasis on identifying drivers of MES reprogramming with the hopes of developing novel GBM therapies. 

Besides the acquisition of invasive and migratory phenotypes, MES reprogramming appears to also activate unique metabolic programs to support demands associated with the aforementioned phenotypes [82]. Thus far, the detailed mechanism of how metabolic alteration synergizes with MES reprogramming is poorly understood in GBM. However, there are some studies that have reported the correlation between metabolic alteration and MES reprogramming in GBM. For instance, Su et al. demonstrated that metabolic and subsequent MES reprogramming in GBM occurs through the TGFβ1-mediated upregulation of NADPH oxidases 4 (NOX4) and reactive oxygen species (ROS), leading to downstream overexpression and nuclear accumulation of hypoxia-inducible factor 1α (HIF-1α) [93]. Utilizing a combination of patient GBM xenografts and patient GBM tissues, Talasila et al. analyzed gene expression changes associated with invasive and angiogenic phenotypes in GBM [94]. They observed an angiogenic switch that was highly correlated with MES programming whereby angiogenic xenografts employed higher rates of glycolysis compared with invasive xenografts. They also noted that MES reprogramming was associated with angiogenic switch through the upregulation of transcriptional factors BHLHE40, CEBEP, and STAT3, which employ higher rates of glycolysis. Lastly, malic enzyme (ME2), an enzyme that catalyzes the formation of pyruvate from malic acid, was found to be highly expressed in GBM and its expression was positively correlated with MES reprogramming through upregulation of MES gene markers and downregulation of PN gene markers [95]. ME2 mediated metabolic reprogramming through inhibition of ROS and AMPK phosphorylation and subsequent facilitation of SREBP-1 nuclear localization, leading to ACSS2 lipogenesis. 

### 4.1. Treatment-Induced MES Reprogramming and Clinical Relevance

One of the most significant challenges in the treatment of GBM is the limited durability of clinical response. As already alluded to, heterogeneity in molecular subtypes, as well as a propensity for clonal evolution toward a more aggressive molecular subtype, have therapeutic–prognostic implications. GBM treatment can reprogram GSCs toward an aggressive MES phenotype, leading to enhanced stemness, invasion, and therapeutic resistance. In particular, treatment-induced MES reprogramming is a significant contributor to GBM therapeutic refractoriness to chemotherapy and radiotherapy [79,80,86,96,97,98]. Both radiation therapy and chemotherapy induce MES reprogramming in GBM preclinically and clinically. In order to overcome the challenge of MES reprogramming, a detailed understanding of molecular mechanisms associated with the MES status of GSCs is necessary.

In an attempt to decipher molecular drivers of MES reprogramming in GCSs, Bhat et al. identified a genetic signature associated with MES transition, radiation resistance, and poor GBM outcomes mediated through NF-κB signaling pathway activation [80]. NF-κB signaling activation reprogrammed GSCs toward an MES phenotype that was highly resistant to radiation resistance through upregulation of CD44 [80]. Paradoxically, radiation therapy has been implicated as a propagator of the MES reprogramming of GBM through the activation of critical MES regulators and signature genes [86,96]. In PN GBM mouse models, radiation treatment induced PN to MES reprogramming both genetically and phenotypically [86]. Furthermore, radiation-mediated MES GBMs are generally more invasive and resistant to TMZ [96]. It is now recognized that the radiation-induced upregulation of TGF-β, VEGF, and PDGF promotes tumor invasion and resistance associated with MES reprogramming [99,100]. New insights into the impact of GBM radiation therapy on brain-invasive GBM cells have provided further enlightenment on transcription programs of MES reprogramming in GBM with potential implications for treatment outcomes [97]. The brain-invasive front of GBM represents a region where the safe resection of tumors is not feasible because tumor cells are highly infiltrated into normal brain tissue. Minata and colleagues recently identified two subpopulations of GSCs within the invasive front of GBM patients, consisting of a CD133+ PN subpopulation and a CD109+ MES subpopulation [97]. Upon exposure to ionizing radiation, CD133+ PN GSCs transitioned to CD109+ MES GSCs, suggesting that radiation induces the expression of CD109 [97]. Mechanistically, the radiation-induced expression of CD109 in GCSs leads to downstream activation of the TAZ/YAP axis, resulting in MES reprogramming, brain invasion, and radiation resistance [97]. Hence, CD109 could serve as a therapeutic target for radiation-induced MES reprogramming in GBM. Approaches to targeting radiation-induced reprogramming have focused on master transcriptional regulators such as STAT3 [96,101] and NF-κB [102] pathways. Targeting STAT3 either through a small molecule inhibitor of survivin, YM155 [101] or through the upstream blockade of STAT3 using JAK2 inhibitors (AZD1480 or ruxolitinib) [96] significantly enhanced radiation sensitivity and prevented MES reprogramming. Recently, it was reported that activation of adhesion G-protein-coupled receptor G1 (GPR56/ADGRG1) could abrogate NF-κB pathway-mediated MES reprogramming in GBM [102].

MES reprogramming is equally a challenge to GBM alkylator chemotherapy where transcription factors such as the forkhead box protein O1 (FOXO1) drive MES resistance reprogramming in GBM to alkylators [103]. TMZ is generally the first-line therapy in GBM administered concurrently with radiotherapy followed by adjuvant TMZ. Given its DNA alkylating mechanism, TMZ treatment leads to a hypermutated and MES phenotype, especially upon tumor recurrence, and further studies evaluating the genetic and phenotypic changes associated with the evolution of TMZ resistance in GBM interestingly revealed the acquisition of an MES gene signature as part of the evolution of GBM cells toward TMZ resistance [104,105,106,107]. In GBM, there are sub-populations of proliferative as well as quiescent GSCs. Not surprisingly, quiescent GSCs are highly refractory to anti-proliferative therapy with TMZ and harbor a very strong TGF-β and HIF1α transgene MES signature [106]. Recent findings have revealed that several key transcriptional pathways play crucial roles in MES reprogramming and TMZ resistance. For instance, FOXO1 affects multiple MES marker genes’ expression and further positively induces TMZ and CDDP (Cisplatin) resistance [103], while STAT3 and NF-κB could induce an immunosuppressive environment associated with TAMs dependent on mTOR activity [108]. 

Beyond alkylating chemotherapy agents, anti-angiogenic agents have been employed as second-line agents in GBM therapy but without any significant impact on overall survival [109,110,111]. Emerging data have implicated anti-angiogenic therapy in promoting GBM tumor hypoxia and MES reprogramming [83,84,112]. Hence, anti-angiogenic therapy failures are often associated with markedly invasive and resistant GBM at recurrence. The MES reprogramming propensity of standard first-line and second-line GBM therapies significantly underscores the urgent need for new treatments that could either prevent or treat MES reprogramming. 

### 4.2. Heterogenous Tumor Microenvironment

The GBM tumor microenvironment (TME) is another major contributor to malignant reprogramming in GBM (Figure 2). Cellular and molecular heterogeneities are highly featured within the TME in GBM (Table 1). Furthermore, distinct histological and MRI-defined regions in GBM with unique cellular compositions and transcriptional programs contribute to intra-tumoral heterogeneity [113] (Table 1). The well-recognized MRI distinct regions include the central necrotic zone, the tumor-enhancing zone, and the peri-tumoral flair or edema region [114,115,116]. The central necrotic region is highly hypoxic and hypocellular with respect to tumor cells, while the tumor-enhancing region is highly angiogenic and hypercellular [114,115,116]. The peri-tumoral flair region harbors brain-invasive GBM cells. Furthermore, histopathologic correlates of the TME include perinecrotic/pseudopalisading regions, the tumor core, and brain-invasive regions. Additional insights into the unique transcriptional programs within each TME niche have emerged through meticulous assessment of laser-microdissected GBM patient tissues [117]. The perinecrotic/pseudopalisading regions of GBM are highly MES and are characterized by HIF1α signaling, TNF-α signaling, and immune response gene enrichment signatures [117]. Not surprisingly, hypoxia is a crucial feature of the perinecrotic/pseudopalisading regions of GBM, and a critical facilitator of both GSC proliferation and angiogenesis [118,119]. While the complete mechanistic underpinnings of the hypoxia-mediated malignant reprogramming of GBM are not fully elucidated, there is mounting evidence that the TME is a critical facilitator. One proposed mechanism of the hypoxia-mediated malignant reprogramming of GBM via the direct activation of pro-angiogenic genes and the subsequent recruitment of inflammatory cells [118,120]. HIF1α signaling secondary to hypoxia has a significant impact on GBM cells. For instance, it has been demonstrated that activation of the HIF1α-ZEB1 axis contributes to GBM invasion and MES reprogramming [121]. Furthermore, genetic silencing or pharmacological inhibition of HIF1α effectively reversed hypoxia-mediated MES reprogramming [121]. Another postulated mechanism of hypoxia-mediated MES reprogramming involves the EPHB2-HIF2α-paxillin signaling axis [122]. HIF2α is required for the stabilization of the tyrosine kinase receptor (TKR) EPHB2 and promotes MES reprogramming by phosphorylating paxillin and focal adhesion kinase (FAK) [122]. Hence, HIF1α- and HIF2α-related mechanisms have MES reprogramming implications for GBM cells within the hypoxic tumor microenvironment. Interestingly, it is now recognized that interactions between normal glial cells with tumor cells can create hypoxia adaptation synergies for tumor cells. For instance, it is postulated that astrocytes within the hypoxic microenvironment release the cytokine CCL20 and upregulate HIF1α in an NF-κB signaling-dependent manner, thereby creating hypoxia adaptation for GBM [123]. Interestingly, the MES reprogramming of GBM cells shares similar genetic signatures with astrocyte reactivity [124]. Collectively, these observations indicate that TME factors cooperate with each other to form interaction networks that promote MES reprogramming in GBM (Figure 2).

Tumor-associated immune cells represent a critical component of the TME. There is mounting evidence that immune-mediated mechanisms are associated with MES reprogramming [80,125,126]. MES subtype GBMs are highly characterized by pro-inflammatory and immunosuppressive genetic profiles [125,126,127,128,129]. Furthermore, tumor infiltrative T lymphocytes are highly represented in MES GBM compared to other GBM molecular subtypes, confirming an immune propensity in MES reprogramming [125,126,130,131]. Recent findings point out that MES-like states may be associated with T cell activation [132]. It is now apparent that of all T cell types, CD8+ T cells are the most represented in MES GBM [127]. Besides lymphoid infiltration, there is mounting evidence of myeloid infiltration into the GBM TME [133,134]. Chemokines and cytokines secreted by GBM cells within the hypoxic niche can activate and recruit TAMs in the TME [135]. MES master transcription regulators such as STAT3 and NF-κB have been implicated as contributors to the immunosuppressive environment associated with TAMs [108]. TAMs as well as microglia can promote hypoxia-induced neovascularization through the release of VEGF and CXC-chemokine ligand 2 (CXCL2) into the TME [136]. TAMs and microglia also express TNF-α, TGF-β, and MMP9, which facilitate the MES reprogramming of GBM cells [80,128,137]. Specifically, the secretion of extracellular matrix remodeling factors along with pro-angiogenic and anti-inflammatory cytokines contributes to an aggressive MES tumor phenotype. Hence, MES GBMs are most commonly associated with macrophage/microglia infiltration and necrosis [51,77,138]. Collectively, interactions between the immune components of TME and GBM cells enhance the adaptive fitness of tumor cells within the hypoxic niche through MES reprogramming (Figure 2). Such interactions can provide valuable insights into the complex cellular and molecular interplay with the TME and may also yield innovative therapeutic targets [139,140].

### 4.3. Key Regulators, Pathways, and Clinical Targets in MES Reprogramming

A deeper understanding of the mechanistic underpinnings related to transcriptional regulators of MES reprogramming in GBM is essential in addressing the unmet need for novel GBM therapeutics (Figure 3). The master regulator of the MES state has been extensively studied in several cancers, including GBM, whereby NF-κB has emerged as a critical regulator of the malignant reprogramming of cancer stem cells [141,142] (Figure 3). NF-κB impacts both tumor cells and the TME. In tumor cells, NF-κB promotes the expression of MES-like markers, while within the TME, NF-κB induces the expression of various pro-inflammatory genes, including those encoding cytokines and chemokines [125,143,144]. The combination of MES expression and a highly proinflammatory TME accounts for the therapeutic resistance of GBM. Besides the direct induction of MES markers in GBM, NF-κB signaling could indirectly mediate MES reprogramming through crosstalk with other regulators including STAT3 and HIF1α [123,145]. Similar to NF-κB, STAT3 exerts transcriptional regulation of both GBM cells and the TME. In GBM cells, co-transcriptional synergistic activation of both STAT3 and C/EBPβ is necessary for MES transformation [146] (Figure 3). As master regulators of MES reprogramming, activation of STAT3 and C/EBPβ induced the transcription of MES genes in GSCs, while suppression of STAT3 and C/EBPβ abrogated the MES gene profile and phenotype. Furthermore, within the GBM TME, activation of both STAT3 and C/EBPβ propagated tumor necrosis and hypoxia [79,138]. Moreover, as previously alluded to, TAMs are significantly abundant in MES GBM, and it now appears that the modulation of STAT3 transcriptional activity in TAMs is a basis for TAM-mediated MES reprogramming in GBM [137]. Besides TAMs, microglia represent another important closely related TME component that modulates GBM cell transcriptional fate. Mechanistically, microglia facilitate TME immunosuppression, tumor immune evasion, and tumor MES transition through the mTOR-dependent regulation of STAT3 and NF-κB [108]. Recently, TAZ, the transcriptional activator with PDZ-binding motif was identified as an MES-related network inducer, whose activity was correlated with GSC invasion and self-renewal (Figure 3). Mechanistically, TAZ forms a complex with the transcriptional enhanced associate domain (TEAD), thereby facilitating the recruitment of TAZ to MES gene promoters [147]. Similar to STAT3 and C/EBPβ, TAZ activity promotes GBM tumor necrosis, which also propagates MES reprogramming and stemness [148]. TAZ can also impact MES reprogramming through its downstream interactions with the Hippo signaling pathway through co-activation of the pathway with Yes-associated protein (YAP) [149]. Although the TAZ transcriptional program appears to be independent of that of STAT3-C/EBPβ despite similarities to GBM tumor cell and TME impacts, both transcription programs intersect with NF-κB. It was recently reported that the NF-κB-mediated MES reprogramming and therapeutic resistance in GSCs occurred through the regulation of STAT3, C/EBPβ, and TAZ [80,150]. Collectively, NF-κB in cooperation with other regulators, serves as a critical master regulator of MES reprogramming within the TME in GBM.

The recent identification of several potential targetable molecular biomarkers of MES reprogramming has raised prospects for clinical translation. Our group identified an anti-apoptotic protein, BIRC3 (baculoviral IAP repeat containing 3), as a biomarker for MES GBM and a mediator of hypoxia-driven survival adaptation through HIF1α signaling [151] (Figure 3). BIRC3 was previously reported as a novel driver of therapeutic resistance in GBM [151,152]. The dual role of BIRC3 in apoptosis evasion and MES reprogramming renders BIRC3 a potential biomarker and therapeutic target for MES GBM that could synergize with cytotoxic chemotherapy. The enzyme transglutaminase 2 (TGM2) is another reported biomarker of the peri-necrotic hypoxic region of GBM [153]. TGM2 has been implicated as a driver of GSC MES reprogramming through the activation of key transcriptional factors including C/EBPβ, TAZ, and STAT3, suggesting that it could be a potential therapeutic target for MES reprogramming [153]. Another reported MES biomarker and potential therapeutic target is S100A4, a gene encoding a small calcium-binding protein that interacts with other key regulators such as p53 [154]. S100A4 has been identified as a critical regulator of GSC self-renewal as well as a reporter of MES reprogramming through the downstream regulation of key transcriptional factors such as SNAIL2 and ZEB1 [154]. Furthermore, the neurotrophic factor prosaposin (PSAP), a conserved glycoprotein that promotes GBM migration/invasion and MES reprogramming via the TGF-β1/SMAD signaling pathway, has been reported as a novel targetable MES biomarker [155] (Figure 3).

There is mounting evidence that the regulatory activities of certain long noncoding RNAs (lncRNAs) contribute to MES reprogramming in GBM through the upregulation of MES genetic markers and MES phenotypes. A novel lncRNA, TALNEC2, was recently reported to be highly expressed in GBM and identified as a regulator of cell proliferation and MES transformation [156]. Silencing of TALNEC2 successfully attenuates both GSC self-renewal and MES reprogramming, leading to radiation sensitivity both in vitro and in vivo [156]. Cooperative interactions between lncRNAs, microRNAs (miRNAs), and other key regulators of MES reprogramming such as the ZEB signaling axis exist in GBM [157,158]. For instance, the lncRNA LINC0051 regulates and promotes MES reprogramming in GBM through the LINC00511/miR-524-5p/ZEB1 signaling axis [157]. Furthermore, there is supportive evidence that through TGF-β activation, ZEB1 could also upregulate another lncRNA, LINC00645, to mediate MES reprogramming through the LINC00645/miR-205-3p/ZEB1 signaling axis [158]. Interestingly, lncRNAs can also suppress MES reprogramming. lncRNA LINC00599 functions as a tumor suppressor in GBM, whereby the expression of LINC00599 significantly attenuates GBM MES reprogramming and tumor aggressiveness [159]. 

Besides lncRNAs, microRNAs have been reported to play an important role in modulating MES reprogramming in GBM. For instance, miR-181c was found to be downregulated in GBM, and the overexpression of miR-181c inhibits TGF-β signaling and further suppresses tumor cell invasion and MES reprogramming [160]. Specifically, miR-181c inhibits TGF-β signaling by downregulating TGFBR1, TGFBR2, and TGFBRAP1 expressions. Recently, Zhang et al. analyzed multiple GBM databases including the TCGA, GSE16011, and Rembrandt and reported that miR-95 and miR-223 have opposing modulatory impacts on MES reprogramming in GBM [161]. Overexpression of miR-95 suppressed MES reprogramming while overexpression of miR-223 facilitated MES reprogramming. The functional correlation between MES reprogramming and miR-223 was further confirmed in a study by Huang et al., in which they showed that the inhibition of the miR-223-PAX6 axis suppressed cell invasion and improved chemotherapy sensitivity [162]. MiR-96 was recently identified as a tumor suppressor and potential therapeutic agent that antagonizes MES reprogramming in GBM through the downregulation of AEG-1 [163]. MiR-101-3p was also found to be a negative regulator of MES reprogramming through the inhibition of TRIM44 signaling [164].

Epigenetic mechanisms have also been implicated in malignant reprogramming in GBM [165,166,167,168]. Interestingly, the inhibition of HDAC6, a histone deacetylase was found to attenuate and also reverse MES signature gene reprogramming in GBM [165]. Histone methyltransferases represent another class of histone modifiers that may play a role in GBM MES reprogramming through the promoter methylation silencing of target genes. For instance, the suppression of H3K27 methylation by enhancer of zeste homolog 2 (EZH2), a histone lysine N-methyltransferase enzyme, reverses MES reprogramming through the upregulation of EZH2 target genes and the downregulation of MES markers [168] (Figure 3). Furthermore, interactions between downstream miRNA targets of EZH2 and key master regulators of MES reprogramming such as TGF-β signaling in GBM have been reported [166]. In particular, EZH2 has been identified as a regulator of the miR-490-3p/TGIF2/TGFBR1 signaling axis [166]. Collectively, histone modifications represent a common phenomenon and may confer therapeutic vulnerability for targeting malignant reprogramming in GBM.

It is worth noting that there are ongoing efforts to identify and target novel critical drivers of MES reprogramming. Understanding and targeting the mechanistic underpinnings of key MES drivers are essential for improving GBM therapeutic outcomes. 

### 4.4. Recent and Potential Therapies Targeting MES Reprogramming

MES reprogramming is a very complicated and challenging phenomenon in GBM with an unmet need for innovative therapies. Currently, there are no effective clinical therapies to treat or prevent MES reprogramming. However, several promising preclinical and clinical agents have been explored and repurposed as anti-MES therapies. Ideally, these agents should have excellent blood–brain barrier penetrance and synergize with standard GBM therapy. 

Recently, paeoniflorin, a natural anti-cancer compound that has been widely studied both in the preclinical and clinical settings, was found to inhibit MES reprogramming and angiogenesis in GBM [169]. Paeoniflorin was first identified as an anti-inflammatory and anti-oxidative drug and later noted to exhibit anti-cancer effects through the induction of apoptosis. In this study, it was demonstrated that paeoniflorin can activate autophagy, promote c-Met degradation via K63-linked polyubiquitination, and further inhibit MES reprogramming and angiogenesis in GBM. 

As previously mentioned, TGF-β is one of the key growth factors that triggers MES reprogramming and angiogenesis in many cancers including GBM. Several anti-TGF-β pharmacologic targeting strategies have been evaluated. Pirfenidone, an anti-fibrosis FDA-approved agent, was reported to inhibit TGF-β expression in malignant glioma cells [170]. Similarly, quetiapine, an FDA-approved anti-psychotic agent, was also reported to inhibit MES reprogramming in a RANKL-TGF-β dependent manner [171]. GBM tumor cells can secret RANKL into TME and increase tumor cell motility to surrounding non-malignant cells, such as astrocytes, and further induce these surrounding cells to secrete TGF-β which in turn reprograms GBM cells to the MES-like invasive type [171]. Thus, combination treatment with quetiapine and pirfenidone may undermine RANKL/TGF-β signaling and interaction between GBM cells and surrounding cells, which can further suppress MES reprogramming [170,171]. Another FDA-approved agent with anti-TGF-β activity is the anti-diabetic agent metformin [172,173]. Metformin was found to inhibit both MES reprogramming and stem-like properties in GBM through TGF-β and AKT/mTOR pathways [174,175]. Given the critical role of TGF-β in MES reprogramming, synergies between standard GBM therapies and the inhibition of TGF-β through pirfenidone, quetiapine, and metformin merits further investigation. 

## 5. Conclusions

Despite significant advances in and enlightenment on the genetic and epigenetic landscapes of GBM, there has been limited progress in improving outcomes for patients afflicted with this very lethal cancer. The interplay between the GSC tumor niche and the TME has emerged as the critical determinant for the therapeutic refractoriness of GBM. Hence, several critical challenges related to the tumor niche as well as the TME niche have to be simultaneously addressed to positively impact therapeutic outcomes. Within the tumor niche, the cellular and molecular heterogeneity of GSC subpopulations modulate clonal adaptation to therapy, leading to malignant reprogramming and therapeutic resistance. Furthermore, the immunosuppressive cellular components of the TME perpetuate malignant reprogramming of the GSC niche. Therefore, the development of therapeutic strategies that prevent clonal adaptation within the GSC niche is highly essential. 

In this review, we have presented an overview of potential therapeutic targets associated with signaling nodes and master regulators of malignant reprogramming. Methodical assessment of potential therapeutic targets could be accomplished through window-of-opportunity clinical trials in recurrent GBM patients undergoing standard-of-care salvage surgery, whereby resected tumor tissue can be analyzed for both drug penetrance and drug target engagement. Hence, developing and establishing the brain-penetrance profile of novel anti-malignant reprogramming therapies should be the objective of future efforts. 

## Figures and Tables

**Figure 1 cells-13-00942-f001:**
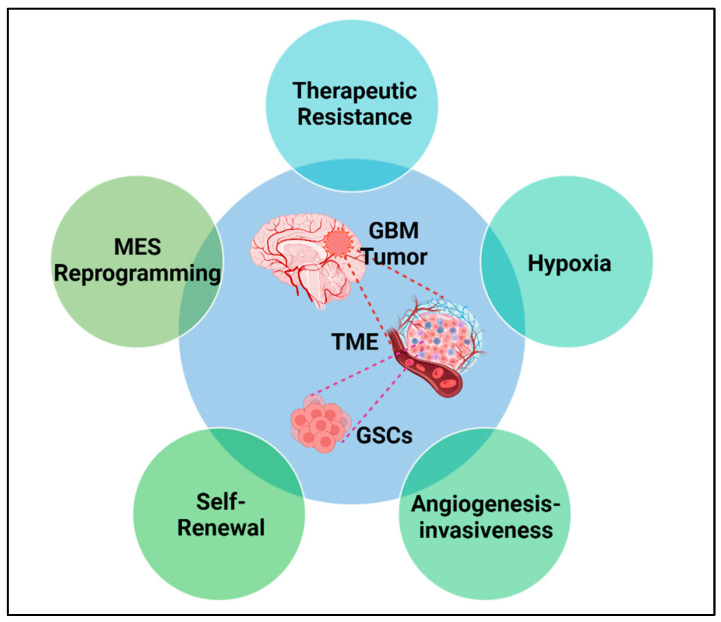
Malignant reprogramming of GSCs in GBM. Schematic representation of the impacts of GSCs on GBM tumor propagation. GSCs contribute to therapeutic resistance, the hypoxic microenvironment, MES reprogramming, tumor cell self-renewal, angiogenesis, and tumor invasion.

**Figure 2 cells-13-00942-f002:**
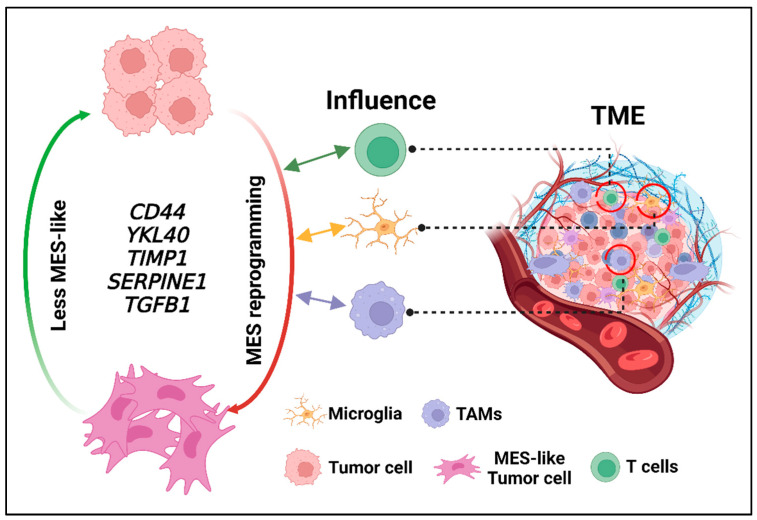
The impact of the GBM tumor microenvironment on MES reprogramming. Representative mechanisms underlying MES reprogramming in the GBM tumor microenvironment. Different cell types from the GBM tumor microenvironment including T cells, tumor-associated macrophages (TAMs), and microglia can interact with GBM tumor cells and further impact GBM cell MES reprogramming. Such MES reprogramming can be demonstrated by specific gene markers including *CD44*, *YKL40*, *TIMP1*, *SERPINE1*, and *TGFB1*.

**Figure 3 cells-13-00942-f003:**
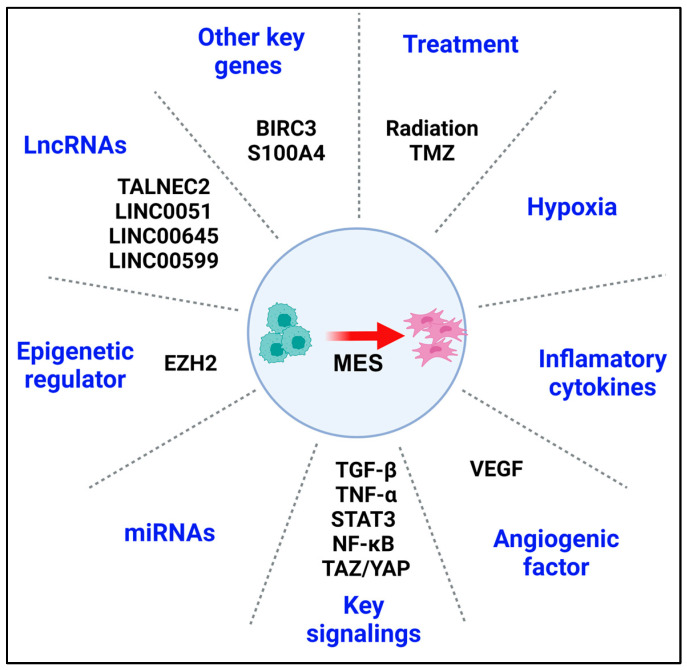
Key factors impacting MES reprogramming. Schematic summary of the most important gene/protein regulators as well as biological events/interventions involved in GBM MES reprogramming. (1) Therapeutic treatment (TMZ, radiation, etc.): treatment can reprogram GSCs toward an aggressive MES phenotype, leading to enhanced stemness, invasion, and therapeutic resistance. (2) Hypoxic environment: a hypoxic TME can mediate malignant reprogramming of GBM through the direct activation of pro-angiogenetic genes and recruitment of inflammatory cells. (3) Inflammatory-related cytokines: these cytokines along with the extracellular matrix contribute toward an aggressive MES tumor phenotype. (4) Epigenetic regulator (EZH2): EZH2-mediated histone methylation plays an important role in the regulation of the expression levels of multiple MES marker and regulator genes. (5) LncRNAs, and (6) miRNAs: these two kinds of non-coding RNAs contribute to GBM MES reprogramming through the regulation of key transcriptional factors, such as ZEB1. (7) Key signaling pathways (TGF-β, TNF-α, STAT3, NF-κB, and TAZ/YAP) and (8) angiogenic factor signaling (VEGF): these key signaling pathways are the master regulator of malignancy, and they frequently interact/collaborate with the hypoxic TME and therapeutic treatment to promote MES reprogramming. (9) Some other key genes (BIRC3 and S100A4): these genes are also identified as the driver of MES reprogramming, through activation of key transcriptional factors including C/EBPβ, TAZ, and STAT3.

**Table 1 cells-13-00942-t001:** Gene expression profiles in different regions of GBM tumors. The overview of the GBM regions reveals distinct patterns and enrichment of specific molecular profiles. The enriched gene profiles were analyzed during the Ivy Glioblastoma Atlas Project.

Tumor Region	Leading Edge and Infiltrating Tumor	Perinecrotic Zone	Pseudopalisading around Necrosis	Hyperplastic Blood Vessels in Cellular Tumor	Microvascular Proliferation
Top 10 Gene Expression Profile	VSNL1	PI3	IL8	COL3A1	ESM1
	CCK	IL8	VEGFA	LOC100506027	COL3A1
	SNAP25	CCL20	HILPDA	LUM	IBSP
	GABRA1	SLPI	NDRG1	COL1A1	CRIP1
	CRYM	SAA1	ADM	ACTG2	LOC100506027
	GNG3	PTX3	CA9	ESM1	HIGD1B
	SYT1	SAA2	CA12	ACTA2	RGS5
	NEFL	TREM1	ANGPTL4	COL6A3	ITGA1
	SYNPR	CHI3L1	HK2	COL1A2	OR51E1
	GABRA2	MMP7	CHI3L1	DCN	MMP9

## Data Availability

Not applicable.

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
