# Peer review of "The Role of Mesenchymal Reprogramming in Malignant Clonal Evolution and Intra-Tumoral Heterogeneity in Glioblastoma"

_cells, 2024, doi:10.3390/cells13110942_

Round 1

Reviewer 1 Report

Comments and Suggestions for Authors

Overall, this is a very interesting and well written review on glioblastoma’s intra-tumoral molecular and cellular heterogeneity.

Major concerns: none.

Minor concerns:

1)    I understand that it is difficult to fit everything in one figure, but the authors should consider revamping Figure 3, if possible.  The text does a much better job at describing the pathways, the genes, and the clinical targets involved in MES reprogramming. The figure should clarify which ones are genes and which ones are pathways. Same for the cytokines. Are these cytokines (e.g., TGF- β, TNF-α) released in the TME? Most likely. It is not at RNA level.

2)    On a similar note, the authors should summarize the factors that impact MES reprogramming rather than saying such factors have been presented. A few sentences highlighting the beautiful findings would make this review much stronger. 

Author Response

Reviewer 1:

  1. I understand that it is difficult to fit everything in one figure, but the authors should consider revamping Figure 3, if possible.  The text does a much better job at describing the pathways, the genes, and the clinical targets involved in MES reprogramming. The figure should clarify which ones are genes and which ones are pathways. Same for the cytokines. Are these cytokines (e.g., TGF- β, TNF-α) released in the TME? Most likely. It is not at RNA level.

R: Thank you very much for your nice suggestion. We have revised our figure. Please see the new Figure 3.

  1. On a similar note, the authors should summarize the factors that impact MES reprogramming rather than saying such factors have been presented. A few sentences highlighting the beautiful findings would make this review much stronger.

R: Thank you for this great advice. We rewrote the figure legend and described related factors in the figure. Please see line 428-line 443.

Reviewer 2 Report

Comments and Suggestions for Authors

The article titled "The Role of Mesenchymal Reprogramming on Malignant Clonal Evolution and Intra-tumoral Heterogeneity in Glioblastoma" presented by Wu et al. is an interesting review that addresses the role of GBM heterogeneity from the intratumoral perspective and its relationship with the tumor microenvironment, as well as the molecular changes leading to therapeutic resistance.

Some considerations that would help improve or focus attention on certain aspects that I believe were superficially addressed:

  1. Please revise the first sentence of the introduction; it mentions a definition of GBM that is not correct, as it is not solely defined by the IDH WT state.
  2. As the authors have mentioned the contribution of genomic profiles in different intratumoral regions of GBM, the aspect of cellular architecture heterogeneity should be described with more support from assays demonstrating it, such as spatial transcriptomic studies.
  3. This statement is incorrect: "Between the outer angiogenic zone and the inner hypoxic zone, lies the perinecrotic zone that contains a significant proportion of highly proliferative GBM stem cells (GSCs)." Please review and support it with bibliography.
  4. The article would be enriched by presenting a diagram of the different regions of GBM and their relationship with the presented genomic profiles.
  5. The following statement requires bibliographic support, otherwise, it appears too speculative: "It became evident that all GBM tumors have PN subpopulations, and that it was the variance of PN subpopulations relative to the other molecular subtypes that impacts on survival. MES subpopulations are highly resistant to therapy and confer a dismal survival compared to other subtypes."
  6. Similarly, with the following statement: "PN towards MES reprogramming represents the most common molecular subtype transition whereby PN genes are down-regulated, and MES genes are upregulated."

  7. In the "Heterogenous Tumor Microenvironment" section where the contribution of other non-tumoral populations present in the microenvironment is mentioned, it would be relevant to cite some references that have addressed this aspect: https://doi.org/10.7554/eLife.52176; https://doi.org/10.3390/brainsci13040542

  8. The section "4.3. Key regulators, pathways, and clinical targets in MES reprogramming" could be enriched by adding contributions related to other non-coding RNAs such as MicroRNA. Please review the literature on this topic. For example: https://doi.org/10.1016/j.bbrc.2015.12.021

Author Response

Reviewer 2:

  1. Please revise the first sentence of the introduction; it mentions a definition of GBM that is not correct, as it is not solely defined by the IDH WT state.

R: Thank you very much for pointing out this. We have removed this controversial statement.

  1. As the authors have mentioned the contribution of genomic profiles in different intratumoral regions of GBM, the aspect of cellular architecture heterogeneity should be described with more support from assays demonstrating it, such as spatial transcriptomic studies.

R: We appreciate this comment. We have added a paragraph and summarized more assays to support cellular architecture heterogeneity in GBM, including scRNA-seq, Spatial transcriptomics. Please see line 136-line 160.

  1. This statement is incorrect: "Between the outer angiogenic zone and the inner hypoxic zone, lies the perinecrotic zone that contains a significant proportion of highly proliferative GBM stem cells (GSCs)." Please review and support it with bibliography.

R: Thank you for the suggestion. We have changed this sentence to a more accurate statement and cited some more references to support it. Please see line 75-line78, line 82-line 84, references 32, 33, 34, 39.

  1. The article would be enriched by presenting a diagram of the different regions of GBM and their relationship with the presented genomic profiles.

R: We appreciate such great idea. We have presented a new table (Table 1) to indicate top 10 genes expression in different GBM tumor regions. Please see Table 1.

  1. The following statement requires bibliographic support, otherwise, it appears too speculative: "It became evident that all GBM tumors have PN subpopulations, and that it was the variance of PN subpopulations relative to the other molecular subtypes that impacts on survival. MES subpopulations are highly resistant to therapy and confer a dismal survival compared to other subtypes."

R: Thank you very much for the comment. We have added a reference to support this point. Please see line 195, reference 27.

  1. Similarly, with the following statement: "PN towards MES reprogramming represents the most common molecular subtype transition whereby PN genes are down-regulated, and MES genes are upregulated."

R: Thank you very much for pointing this out. We have also added some refs to support this statement. Please see line 218, references 80, 89-91.

  1. In the "Heterogenous Tumor Microenvironment" section where the contribution of other non-tumoral populations present in the microenvironment is mentioned, it would be relevant to cite some references that have addressed this aspect: https://doi.org/10.7554/eLife.52176; https://doi.org/10.3390/brainsci13040542

R: We appreciate this great suggestion. We have cited these several references to give more support to this aspect. Please see line 382-384, references 139, 140.

  1. The section "4.3. Key regulators, pathways, and clinical targets in MES reprogramming" could be enriched by adding contributions related to other non-coding RNAs such as MicroRNA. Please review the literature on this topic. For example: https://doi.org/10.1016/j.bbrc.2015.12.021

R: We thank you for the advice. We have added a new paragraph about the impact of miRNA on the MES reprogramming to support such aspect. Please see line 479- line 493.

Reviewer 3 Report

Comments and Suggestions for Authors

Summary of the Paper: The respected authors have prepared a very timely manuscript focusing on "Mesenchymal Reprogramming on Malignant  Clonal Evolution and Intra-tumoral Heterogeneity in Glioblastoma, They have explained Glioblastoma (GBM) as the most common and fatal type of adult brain cancer, primarily due to its molecular and cellular diversity, leading to treatment resistance and poor prognosis. They mentioned that It exhibits distinct subtypes, with the proneural (PN) subtype having better survival rates than the mesenchymal (MES) subtype, which is highly resistant. GBM stem cells (GSCs) play a crucial role in tumor recurrence and progression, evolving into different subtypes in response to therapy, thus worsening outcomes. The shift of GSCs to the MES subtype under current treatments is a key factor in treatment resistance. Addressing this evolution towards MES through innovative therapies could significantly improve GBM patient survival. In their current review paper they further explore and explain the complexities of MES reprogramming in GBM progression and suggests future strategies to overcome treatment resistance, a critical clinical challenge. Overall, it is a strong review which is supported with strong organization and timely update. They have included innovative figures which made it easy for the reader to follow the story.

Major:

1- It is very important if the respected authors provide details on MES reprograming.

2- For the heterogenity of GBM, how the respected authors link it to common mutations?

3- What is the effect of metabolic switch in the MES reprograming?

4- Please include a section about recent innovation in therapy targeting MES reprograming. 

Author Response

Reviewer 3:

  1. It is very important if the respected authors provide details on MES reprograming.

R: We appreciate this comment. We have added some detail introduction on MES reprogramming process. Please see line 205 – line 216.

  1. For the heterogenity of GBM, how the respected authors link it to common mutations?

R: We appreciate this comment and great suggestion. We have added a paragraph about the relationship of genomic mutations and GBM heterogeneity in section 3. Please see line 110 – line 132. 

  1. What is the effect of metabolic switch in the MES reprograming?

R: Thank you very much for the comment. We have added a paragraph about the metabolic changes/switch during MES reprogramming in section 4. Please see line 226 – line 247. 

  1. Please include a section about recent innovation in therapy targeting MES reprograming.

R: Thank you for this great suggestion. We have included a section about recent therapeutics targeting MES reprogramming. Please see section 4.4, line 512 – line 543.

Round 2

Reviewer 2 Report

Comments and Suggestions for Authors

I believe that the contributions made by the authors have substantially improved the article and corrected some errors. I have no further comments on this matter and consider that the article is now ready for publication.